# Impact of CCD Inhibition on Semi-Quantitative Multiple Allergen Simultaneous Test

**DOI:** 10.3390/diagnostics15222869

**Published:** 2025-11-12

**Authors:** Hyeyoung Lee, Kyungcheol Song, Jiwoo Kim, Jehyun Koo, Jayoung Kim

**Affiliations:** 1Department of Laboratory Medicine, International St. Mary’s Hospital, College of Medicine, Catholic Kwandong University, Incheon 22711, Republic of Korea; shomermaid@catholic.ac.kr (H.L.); rudcjf012@ish.ac.kr (K.S.); freeguy9040@ish.ac.kr (J.K.); andrea5169@ish.ac.kr (J.K.); 2Biomedical Institute of Mycological Resource, International St. Mary’s Hospital, College of Medicine, Catholic Kwandong University, Incheon 22711, Republic of Korea

**Keywords:** allergens, IgE, CCD

## Abstract

**Background/Objectives**: Cross-reactive carbohydrate determinants (CCDs) are a well-recognized source of false-positive results in allergen-specific IgE assays, leading to overestimation of polysensitization. This study aimed to evaluate the impact of CCD inhibition on semi-quantitative multiple allergen simultaneous test (MAST) using the AdvanSure™ AlloScreen Max108 Panel (LG Chem, Seoul, Republic of Korea). **Methods**: We analyzed 65 serum samples positive for anti-CCD IgE (class ≥ 1). A total of 6624 allergen-specific IgE results across foods, pollens, mites, animal epithelia, and fungi were compared before and after CCD-blocker (20 µg) pretreatment. **Results**: After CCD-blocker pretreatment, a total of 1686 allergen-specific IgE results converted from positive to negative out of 2631 positives before inhibition (overall conversion rate 64.1%). The reversion to negative rate declined progressively with increasing MAST class (*p* < 0.0001 for trend). By allergen group, CCD-blocker pretreatment reduced IgE positivity from 28.2% to 8.1% in foods, from 70.1% to 23.3% in pollens, and from 41.0% to 29.2% in mites (all *p* < 0.001), whereas animal epithelia and fungi exhibited minimal or no significant inhibition. Notably, while CCD antigens themselves exhibited class-dependent conversion rates, non-CCD allergens did not show significant differences by CCD class. In venom allergens, honey bee and yellow jacket IgE levels decreased significantly, with resolution of double-positivity in 94.1% of cases. In the dose–response analysis, increasing the CCD-blocker concentration from 20 µg to 40 µg yielded additional inhibition in selected pollens and foods, while most other allergens showed minimal incremental change. **Conclusions**: CCD inhibition markedly improves the specificity of MAST by reducing false-positive results, particularly in food and pollen allergens.

## 1. Introduction

The measurement of serum allergen-specific immunoglobulin E (sIgE) plays a pivotal role in diagnosing and managing IgE-mediated hypersensitivity reactions and allergic disorders [1]. Multiple allergen simultaneous test (MAST) has been widely adopted, offering benefits such as minimal sample volume, rapid turnaround time, and cost-effectiveness across a broad range of sIgE targets [2]. These assays enable the simultaneous screening of multiple allergens and report semiquantitative class results (ranging from 0 to 6), thereby providing practical information for clinical interpretation [3].

Glycosylation is a common post-translational modification affecting protein function and stability [4]. Cross-reactive carbohydrate determinants (CCDs) are carbohydrate structures attached to amino acids during the glycosylation typically characterized by core α1, 3-fucose and β1, 2-xylose on N-glycans. Although CCDs often induce anti-CCD IgE antibodies, these antibodies usually lack clinical relevance because they do not elicit effector cell activation, leading instead to cross-reactivity and false-positive results in allergy testing [5,6,7]. CCDs are widely distributed in allergen extracts from pollen, foods, insects, and latex [8]. Previous studies have shown that CCD-specific IgE is detectable in approximately 20% of allergic patients [6] and may reach up to 35% in younger populations [9]. Because CCD-sIgE often results in false-positive reactions, patients may be incorrectly advised to avoid certain allergens. In some cases, this may even lead to unnecessary or inappropriate treatment [10]. The CCD-blocker competitively inhibits these non-clinically relevant IgE bindings, allowing only true allergen-specific IgE directed against protein epitopes to be detected [5,6]. Therefore, reliable identification of CCD-specific IgE is essential to improve diagnostic accuracy and guide appropriate management of allergic diseases [11]. Recent studies have emphasized the clinical consequences of CCD-driven false positivity in practical allergy diagnostics, showing that inhibition of CCD binding improves the correlation between in vitro sIgE results [4,12].

MAST are widely used in clinical laboratories and establishing population-specific reference ranges and standardized interpretative algorithms is necessary to ensure consistency and clinical reliability of test results [2]. However, systematic studies on the prevalence and clinical relevance of CCD-sIgE remain limited, and standardized procedures or interpretative guidelines for CCD inhibition testing have not yet been established. Given the clinical impact of CCD-sIgE on allergy diagnostics, this study aimed to systematically evaluate CCD interference in MAST by comparing sIgE profiles before and after CCD-blocker pretreatment and assessing its diagnostic utility.

## 2. Materials and Methods

### 2.1. Study Design

From November 2021 to November 2023, we analyzed residual serum samples from patients who underwent allergen-specific IgE testing using the AdvanSure™ AlloScreen Max108 Panel (LG Chem, Seoul, Republic of Korea) in International St. Mary’s hospital. A total of 65 serum samples from patients with CCD specific IgE antibody levels of class 1 or higher were included, comprising 6624 allergen-specific IgE results. Samples were excluded if microbial contamination was suspected, the volume was insufficient for testing, the container was damaged or unlabeled, or if storage was inappropriate or unverifiable. This study was approved by the Institutional Review Board of International St. Mary’s Hospital, Catholic Kwandong University (IS22ESSE0021).

### 2.2. Allergen Specific IgE Assay

Serum allergen-specific IgE was measured using the AdvanSure™ AlloScreen Max108 Panel (LG Chem, Republic of Korea), a multiplex enzyme immunoassay performed on an immunoblotting platform [13]. A result ≥ 0.35 kU/L was considered positive. AlloScreen assay applied the same semiquantitative classification scale, expressed in kU/L: class 0 (0.00–0.34), class 1 (0.35–0.69), class 2 (0.70–3.49), class 3 (3.50–17.49), class 4 (17.50–49.99), class 5 (50.00–99.99), and class 6 (≥100) [14].

### 2.3. Inhibition of Anti-CCD IgE

To reduce nonspecific reactivity to CCD antigens, the ProGlycAn CCD-blocker (ProGlycAn, Vienna, Austria) was employed. The CCD-blocker functions by competitively binding to CCD-specific antibodies present in human serum, thereby preventing their interaction with CCD epitopes on allergens. Patient serum (100 µL) was mixed with 2 µL of the ProGlycAn CCD-blocker according to the manufacturer’s recommendation and previous validation studies [12,15], resulting in a final concentration of 20 μg/mL.

### 2.4. Dose–Response Evaluation of CCD-Blocker Pretreatment

Residual serum samples exhibiting polysensitization with pre-specified as ≥10 total positive allergens and ≥5 positives among pollens and foods were selected. Each serum sample was split into aliquots for pretreatment with 20 µg CCD blocker, and 40 µg CCD blocker. All aliquots from the same specimen were processed in a single run on the AdvanSure™ AlloScreen MAST immunoblot, and results were reported on the semi-quantitative class scale (class 0–6). The inhibitory effect was assessed by identifying allergens that showed additional reversion to negative (class 0) or further class downgrades when the CCD-blocker concentration was increased from 20 µg to 40 µg.

### 2.5. Statistical Analysis

Statistical analyses were performed using GraphPad Prism version 10.3.0 (GraphPad Software, San Diego, CA, USA). Continuous variables are expressed as mean ± standard deviation (SD) or median with interquartile range (IQR), as appropriate. Categorical variables were compared using the chi-square test or Fisher’s exact test. Trends in reversion to negative rates across CCD or allergen-specific IgE classes were evaluated using the Chi-square test for trend. A two-sided *p* value < 0.05 was considered statistically significant.

## 3. Results

### 3.1. Overall Changes in Allergen-Specific IgE Positivity After CCD-Blocker Pretreatment

A total of 65 patients were included in this study, consisting of 50 males (76.9%) and 15 females (23.1%). Thirteen patients (20.0%) were younger than 18 years, 37 (56.9%) were 18–64 years, and 15 (23.1%) were greater than 65 years (Table 1). In total, 6624 sIgE tests were performed (foods 45.5%, pollens 30.6%, mites 8.7%, animal epithelia 6.8%, fungi 6.6%, latex 1.0%, CCD 1.0%). Among the 2631 results with sIgE levels of class ≥ 1, the overall positivity rate for sIgE was significantly reduced from 39.7% before CCD-blocker treatment to 14.3% after treatment (*p* < 0.0001) (Table 2). By allergen group, significant reductions in allergen-specific IgE positivity were observed in the foods, pollen, and mite groups, decreasing from 28.2%, 70.1%, and 41.0% before CCD-blocker treatment to 8.1%, 23.3%, and 29.2% after treatment, respectively (all *p* < 0.0001). In contrast, no significant reduction in positivity was observed for the animal epithelia or fungal allergen groups. For CCD antigens themselves, the number of positive results decreased markedly from 65 (100%) before inhibition to 9 (13.8%) after CCD-blocker treatment (*p* < 0.0001).

### 3.2. Reversion to Negative Rate by CCD and Allergen Specific IgE Class

To investigate whether the extent of CCD sensitization in each specimen affected the inhibition efficiency, we analyzed reversion to negative rates across all allergens according to the specimen’s CCD antigen class (Figure 1). For CCD antigens, the reversion to negative rate after CCD inhibition was 100% for class 1 and class 2, 88.8% for class 3, 16.7% for class 4, and 0% for class 5 (chi-square test for trend, *p* < 0.0001). In contrast, for total allergens excluding CCD, no significant differences in reversion to negative rates were observed according to CCD class. Similarly, within the pollen and food groups, the proportion of allergens converting to negative did not significantly differ by CCD class.

To evaluate whether the degree of allergen-specific IgE reactivity influences the extent of CCD inhibition, we analyzed reversion to negative rates according to the MAST class of individual sIgE results (Figure 2). The distribution of sIgE results was as follows: class 1 (n = 604), class 2 (n = 1053), class 3 (n = 668), class 4 (n = 213), class 5 (n = 71), and class 6 (n = 22). The proportion reverting to negative declined sharply with increasing MAST class (chi-square test for trend, *p* < 0.0001): 90.7%, 79.5%, 42.1%, 8.9%, 1.4%, and 0% from class 1 to 6, respectively. A similar trend was observed for both pollen (95.7 → 0%) and food allergens (93.2 → 0%) (chi-square test for trend, *p* < 0.0001).

### 3.3. Antigen-Specific Effects

The food panel consisted of 50 allergens, of which 45 were included in the CCD inhibition analysis, yielding a total of 848 sIgE results. Among these, 25 food allergens showed a statistically significant reduction in sIgE positivity after CCD-blocker pretreatment (*p* < 0.05) (Figure 3). Among food allergens, conversion rates exceeded 80% for banana, maize, barley, chestnut, mango, and wheat, whereas peach showed the lowest inhibition (44%).

The pollen panel consisted of 33 allergens, of which 32 were included in the CCD inhibition analysis, yielding a total of 1420 sIgE results with anti-CCD antibodies. After CCD-blocker pretreatment, sIgE positivity was significantly reduced for all pollen allergens except birch (*p* < 0.005). The strongest effects were observed in tree pollens such as hinoki cypress, alder, acacia, and poplar, and in weeds including cocklebur and sycamore, where conversion rates exceeded 80%. Grass pollens showed variable but generally moderate reductions (approximately 50–75%), while birch remained largely unaffected (<25%).

The fungal and mite panel consisted of 17 allergens, of which 14 were included in the CCD inhibition analysis, yielding a total of 247 allergen-specific IgE results with anti-CCD antibodies. Among mite allergens, *Dermatophagoides farinae*, *Dermatophagoides pteronyssinus*, and house dust showed almost no reversion to negative after CCD-blocker pretreatment. Only honey bee and yellow jacket venoms demonstrated a statistically significant reduction in sIgE positivity (*p* < 0.05). CCDs are a well-known cause of double positivity to honey bee and yellow jacket venoms in patients with Hymenoptera allergy. In our study, 17 sera were double-positive before CCD inhibition. After pretreatment, 6 samples (35.3%) became double-negative, 10 (58.8%) turned negative for one venom, and 1 (5.9%) showed no change.

### 3.4. Dose–Response Evaluation Results 

A total of 856 sIgE results from eight polysensitized serum samples were included in the dose–response analysis. Increasing the CCD-blocker concentration from 20 µg to 40 µg produced additional reversion to negative or class downgrades in selected allergens, primarily within the food and pollen groups (Table 3). Among food allergens, tomato showed the greatest incremental inhibition (6/8, 75%), followed by celery (4/8, 50%), and several other foods such as peach and potato (3/8, 37.5%). Moderate additional inhibition (25%) was also observed in barley, rice, wheat, and several other foods, whereas most remaining food allergens exhibited minimal change (≤12.5%). In the pollen group, notable additional inhibition was observed for dandelion and sallow willow (4/8, 50%), as well as acacia, ash, bent grass, English plantain, and pine (3/8, 37.5%). Moderate incremental inhibition (25%) was observed in other tree and weed pollens, while most remaining pollens exhibited minimal change (≤12.5%). In contrast, most non-plant allergens including mites, animal epithelia, fungi, and venoms showed little or no additional inhibition.

## 4. Discussion

In this study, we demonstrated that CCD-blocker pretreatment significantly lowered sIgE positivity in MAST results, most notably for food and pollen allergens. The overall positivity rate decreased from 39.7% to 14.3%, with the strongest inhibition seen in plant-derived allergens. These findings underscore the considerable impact of CCDs on in vitro allergy diagnostics and highlight the importance of addressing this analytical source of false positivity.

The effectiveness of CCD inhibition has been documented across various in vitro allergy diagnostic platforms. Because of the cellulose-based antigen presentation, assays such as ImmunoCAP are especially vulnerable to CCD-driven cross-reactivity, and several reports have shown that CCD-blockers substantially reduce false-positive results [5,16]. Holzweber et al. observed that CCD inhibition improved agreement with both patient history and skin prick testing [9], while subsequent studies demonstrated improved consistency between multiplex IgE assays and clinical histories when CCD-blockers were applied [11,12,15,17]. Our findings extend these observations by systematically evaluating CCD-blocker pretreatment in MAST across more than 6000 allergen-specific IgE results, providing one of the most comprehensive datasets to date.

Importantly, our stratified analysis revealed that inhibition efficiency was strongly dependent on sIgE class rather than CCD class. Reversion to negative rates were over 90% in MAST class 1, nearly 80% in class 2, and 40% in class 3, but dropped sharply to below 10% in class ≥ 4. In contrast, CCD class showed predictive value only for CCD antigens themselves; non-CCD allergens did not display significant differences in conversion according to CCD class. This indicates that allergen-specific IgE class is the primary determinant of inhibition efficiency in clinical practice, whereas CCD class alone provides limited predictive value. This pattern partially differs from previous studies, which suggested that low-to-intermediate CCD-sIgE classes are more susceptible to inhibition, while high-class reactivity persists due to true protein-epitope sensitization [12,16].

Birch pollen and peach consistently showed low conversion rates after CCD inhibition (<50%), suggesting that IgE reactivity to these allergens is more likely attributable to genuine protein epitopes rather than CCDs. Birch contains Bet v 1, a PR-10 protein widely recognized as a clinically relevant allergen [18]. Peach contains Pru p 3, a lipid transfer protein known to cause severe food allergy [19]. Conversely, many other food allergens and tree pollens showed conversion rates exceeding 80%, indicating that CCD interference contributes substantially to false-positive results in these groups.

Mite allergens, including *Dermatophagoides farinae*, *D. pteronyssinus*, and house dust, exhibited almost no inhibition after CCD blocking, aligning with a prior study [15]. Some insect venoms, such as honey bee and yellow jacket, which contain CCDs, have also been reported to show reduced reactivity after CCD inhibition [7,20,21,22]. In our study, 94.1% of double-positive sera were resolved after CCD inhibition, confirming that CCDs are a major contributor to this phenomenon.

According to our findings, a CCD-blocker concentration of 20 µg is optimal for multiplex MAST profiles showing multiple low-to-intermediate positive results (predominantly classes 1–3). When residual positivity remains—particularly for plant-derived allergens—or when the clinical history is inconsistent with laboratory findings, a targeted higher-dose inhibition test at 40 µg may be considered to clarify residual CCD-related reactivity. The standard 20 µg concentration was chosen according to the manufacturer’s recommendation and prior validation studies [12,15]. While a higher dose of the CCD-blocker may enhance inhibition, the observed pattern of pronounced suppression in low to moderate sIgE classes and minimal effect in high classes suggests that nonspecific reduction is unlikely. Nevertheless, confirmatory testing using recombinant, non-glycosylated allergen components would further verify that true protein-epitope recognition is preserved. Because the dose–response evaluation was performed on only eight highly polysensitized sera, this component of the study should be regarded as exploratory rather than definitive evidence regarding optimal dosing. The small sample size reflects the limited availability of sera demonstrating extensive CCD reactivity. Nevertheless, it provided useful preliminary insight, showing that 40 µg offered only modest additional inhibition beyond 20 µg for most allergens. These results suggest that 20 µg is generally sufficient for analytical purposes, while larger, prospectively designed studies will be required to validate the optimal dosing strategy and confirm reproducibility across different allergen groups.

As this was a retrospective, single-center study, extrapolation of our findings to broader patient populations should be approached with caution. The lack of longitudinal clinical correlation also limits direct assessment of clinical relevance. Future multicenter studies integrating analytical and clinical data will be essential to validate and generalize these findings.

This study has several limitations. First, it was conducted at a single institution, which may limit the generalizability of the findings. Second, clinical symptoms and patient histories were not systematically correlated with laboratory results, preventing direct assessment of the clinical relevance of CCD inhibition. Third, because only samples with anti-CCD IgE ≥ class 1 were included, there is an inherent selection bias toward CCD-sensitized individuals. As a result, the overall degree of inhibition and the proportion of reversion to negative observed in our results may be overestimated compared with what would be expected in an unselected allergy-testing population. These findings therefore primarily reflect the analytical interference patterns within CCD-sensitized sera and should be generalized to routine testing populations with caution.

Despite these limitations, our study has notable strengths. It is among the largest to date to systematically analyze CCD-blocker effects in multiplex testing, covering diverse allergen groups with over 6000 individual results. By explicitly stratifying results according to sIgE class and CCD class, we provide new evidence that inhibition efficiency is primarily driven by sIgE class and only secondarily by CCD class. The breadth of allergen groups evaluated—including foods, pollens, mites, fungi, epithelia, and venoms—further enhances the generalizability of our findings to real-world laboratory practice.

This study demonstrates analytical improvement in assay specificity after CCD-blocker pretreatment. While these findings suggest potential diagnostic benefit, further multicenter and clinically validated studies are necessary before routine clinical implementation.

In conclusion, CCD-blocker pretreatment substantially enhances the analytical specificity of multiplex allergen testing by reducing false-positive results, particularly among plant-derived allergens. The inhibitory effect is primarily driven by sIgE class, whereas CCD class shows limited influence. These results highlight the analytical value of CCD inhibition. Further prospective multicenter and clinically validated studies are needed to confirm its clinical applicability and support future diagnostic standardization.

## 5. Conclusions

In summary, CCD-blocker pretreatment markedly improved the analytical specificity of multiplex allergen testing by effectively reducing CCD-related false-positive reactions, particularly among plant-derived allergens such as foods and pollens. The degree of inhibition was closely correlated with sIgE class, showing strong suppression in low-to-intermediate classes and minimal impact at high classes, while CCD classes influenced only CCD antigens themselves. These findings suggest that incorporating CCD inhibition into routine MAST workflows can enhance diagnostic accuracy, preventing misinterpretation of polysensitization. Future multicenter studies integrating clinical histories and outcome-based validation are warranted to establish standardized guidelines for CCD-blocker application and optimize its clinical utility in allergy diagnostics.

## Figures and Tables

**Figure 1 diagnostics-15-02869-f001:**
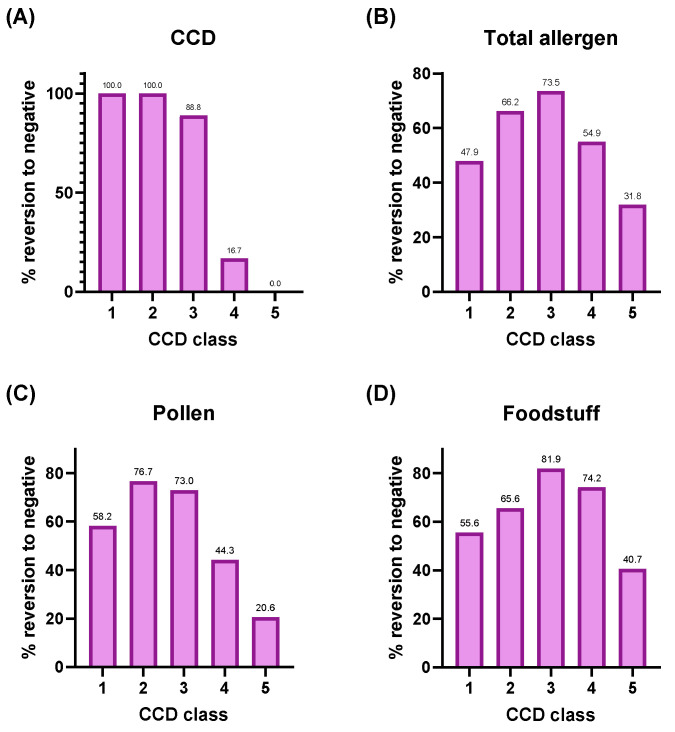
Reversion to negative rates after CCD-blocker pretreatment stratified by CCD class. The inhibitory effect of CCD-blocker pretreatment was evaluated across all allergens according to the CCD antigen class of each specimen. Panels show CCD antigens (**A**), total allergens (**B**), pollens (**C**), and food allergens (**D**). CCD antigens themselves exhibited a clear class-dependent inhibition (chi-square test for trend, *p* < 0.0001), whereas non-CCD allergens showed no significant trend according to CCD class.

**Figure 2 diagnostics-15-02869-f002:**
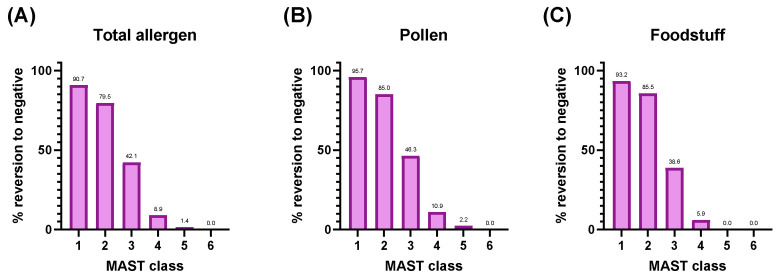
Reversion to negative rates after CCD-blocker pretreatment stratified by allergen-specific IgE class. The inhibitory effect of CCD-blocker pretreatment was analyzed across allergens according to MAST class. Panels represent total allergens (**A**), pollens (**B**), and food allergens (**C**). A distinct class-dependent decline in reversion to negative rates was observed for total allergens, pollens, and food allergens (chi-square test for trend, *p* < 0.0001).

**Figure 3 diagnostics-15-02869-f003:**
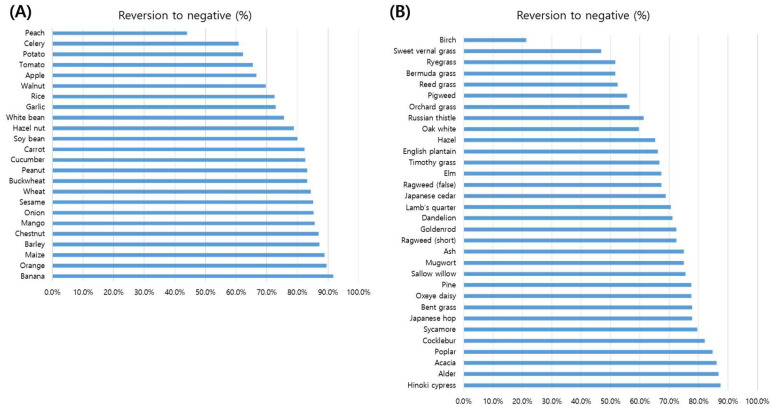
Antigen-specific inhibitory effects of CCD-blocker pretreatment. The inhibitory effects of CCD-blocker pretreatment are shown for food (**A**) and pollen allergens (**B**). (**A**) Food allergens: Significant inhibition was observed across most food antigens, but peach exhibited the lowest conversion rate. (**B**) Pollen allergens: Nearly all pollens demonstrated marked inhibition, except birch pollen.

**Table 1 diagnostics-15-02869-t001:** Clinical characteristics of the study population and distribution of tested allergens.

Characteristics	Number	%
Gender		
Male	50	76.9%
Female	15	23.1%
Age (year)		
<18 years	13	20.0%
18–64 years	37	56.9%
≥65 years	15	23.1%
Tested allergens	6624	
Food stuffs	3011	45.5%
Pollens	2025	30.6%
Mites	575	8.7%
Animal epithelia	450	6.8%
Fungi	434	6.6%
Other (latex)	64	1.0%
CCD	65	1.0%

**Table 2 diagnostics-15-02869-t002:** Positivity rates of allergen-specific IgE before and after CCD-blocker treatment.

Antigen Group	Number	Before CCD Blocker, n (%)	After CCD Blocker, n (%)	*p* Value
Food stuffs	3011	848 (28.2%)	244 (8.1%)	<0.0001
Pollens	2025	1420 (70.1%)	472 (23.3%)	<0.0001
Mites	575	236 (41.0%)	168 (29.2%)	<0.0001
Animal epithelia	450	43 (9.6%)	41 (9.1%)	0.7968
Fungi	434	11 (2.5%)	10 (2.3%)	0.8474
Other (latex)	64	8 (12.5%)	1 (1.6%)	0.0098
CCD	65	65 (100%)	9 (13.8%)	<0.0001
Total	6624	2631 (39.7%)	945 (14.3%)	<0.0001

**Table 3 diagnostics-15-02869-t003:** Dose–response evaluation of CCD-blocker pretreatment (n = 8 sera).

Allergen Group	Allergens	Additional Inhibition (%)
Food	Tomato, Celery	50–75
	Peach, Potato	25–50
	Apple, Barley, Buckwheat, Garlic, Rice, Walnut, Wheat, Carrot, Chestnut, Cucumber, Onion, Sesame, Soybean	<25
Pollens	Dandelion, Sallow willow	50–75
	Acacia, Ash, Bent grass, English plantain, Pine	25–50
	Elm, Goldenrod, Hazel, Lamb’s quarter, Mugwort, Oxeye daisy, Ragweed (short), Russian thistle, Sycamore, Timothy grass, Alder, Birch, Bermuda grass, Cocklebur, Oak white, Orchard grass, Pigweed, Ragweed (false), Reed grass, Ryegrass	<25
Mites/Epithelia/Fungi/Others	CCD	25–50
	House dust, Yellow jacket, *D. farinae*, Cat, Dog, Silkworm	<25

## Data Availability

The data presented in this paper are available on request from the corresponding author.

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
