# Peer review of "Impact of CCD Inhibition on Semi-Quantitative Multiple Allergen Simultaneous Test"

_diagnostics, 2025, doi:10.3390/diagnostics15222869_

Round 1

Reviewer 1 Report

Comments and Suggestions for Authors

This manuscript presents a well-designed and comprehensive evaluation of the effect of CCD-blocker pretreatment on a large dataset of allergen-specific IgE results using the MAST platform. The topic is clinically relevant, as CCD interference is a known issue in allergy diagnostics, and the systematic assessment of its impact across multiple allergen groups is a strength. The study is clearly written, the methods are appropriately described, and the statistical analyses are sound. However, several issues should be addressed before the manuscript is suitable for publication.

  1. The study included only samples with anti-CCD IgE ≥ class 1. While this ensures a CCD-sensitized cohort, it may overestimate the overall impact of CCD interference in a general allergy population. The authors should discuss the potential for selection bias and its implications for generalizability.

  1. In Figures 1 and 2, the term “Grade” for CCD and MAST is not clear. Add an explanation of these terms in the text.

  1. The dose–response evaluation was performed on only eight samples. This small subset limits the generalizability of the findings regarding the incremental benefit of 40 µg CCD-blocker. The authors should either expand this analysis or temper their conclusions regarding optimal dosing.

  1. It is not clear from the text or the methodology where the decision to use 20 µg of the blocking agent is based. I am concerned that it may be an excess and that the decrease in positivity is due to an excess of the blocking agent. You should mention it, discuss it, and, if possible, perform a test with a recombinant protein to demonstrate that its detection is not reduced when using 20-40 µg of the CCD-Blocker.

  1. Table 3 is complex and may benefit from reorganization or simplification for better readability.
  2. The term “negative conversion” is used throughout. Consider using “reversion to negative” or “loss of positivity” for clarity.
  3. The reference list is generally appropriate, but some older citations (e.g., Aalberse et al., 1981) could be complemented with more recent reviews on CCDs and cross-reactivity.
  4. Minor grammar issues. Line 80 (2.2 Allergen..), Line 177, 182 (space on the word sIgE). Lines 193-197 very long sentence. Table 3, farinae must be in italics.

With these revisions, the manuscript will be a strong contribution to the field of allergy diagnostics.

Reviewer 2 Report

Comments and Suggestions for Authors

1- The introduction outlines the main biochemical issue (CCD interference in IgE assays) and situates the MAST platform adequately but could be enhanced by integrating a concise clinical impact statement, prior assay comparison references, and a more explicit justification of study novelty. References are generally appropriate but sparse; the section would benefit from two to three additional citations linking CCD-related false positives to practical allergy diagnostics and illustrating the progression of CCD‑blocker applications.

2-The experimental setup appropriately addresses the analytical question regarding CCD‑blocker effects in multiplex allergy testing and benefits from a large dataset and clear stratification scheme. Nonetheless, the retrospective, single‑center nature and reliance on descriptive metrics limit the generalizability and causal interpretability of results.

 3- The Results section is well organized and quantitatively transparent, allowing readers to follow analytical progression from overall inhibition through stratified analyses. Figures and tables effectively complement the text. However, minor redundancy and limited interpretive commentary somewhat reduce narrative clarity. Condensing repetitive numeric descriptions, enhancing figure references, and inserting brief interpretive statements would make the presentation more concise and reader‑friendly while maintaining statistical rigor.

4- The conclusions accurately reflect the principal analytical findings that CCD‑blocker pretreatment enhances assay specificity and that inhibition is primarily driven by sIgE grade rather than CCD grade. These inferences are well-grounded in the presented data and align with statistical outcomes. However, statements regarding clinical implementation and patient care benefits extend beyond the direct evidence of this single‑center analytical study. The conclusion would be stronger if limited to analytical performance and coupled with a recommendation for future multicenter or clinically validated studies before translation into diagnostic guidelines.

Round 2

Reviewer 2 Report

Comments and Suggestions for Authors

No additional comment